# Sparse2Dense: Learning to Densify 3D Features for 3D Object Detection

**Tianyu Wang[1,2,3], Xiaowei Hu[3,*], Zhengzhe Liu[1], Chi-Wing Fu[1,2]**

[1] The Chinese University of Hong Kong
[2] The Shun Hing Institute of Advanced Engineering
[3] Shanghai AI Laboratory
{wangty,zzliu,cwfu}@cse.cuhk.edu.hk, huxiaowei@pjlab.org.cn

## Abstract

LiDAR-produced point clouds are the major source for most state-of-the-art 3D object detectors. Yet, small, distant, and incomplete objects with sparse or few points are often hard to detect. We present Sparse2Dense, a new framework to efficiently boost 3D detection performance by learning to densify point clouds in latent space. Specifically, we first train a dense point 3D detector (DDet) with a dense point cloud as input and design a sparse point 3D detector (SDet) with a regular point cloud as input. Importantly, we formulate the lightweight plug-in S2D module and the point cloud reconstruction module in SDet to densify 3D features and train SDet to produce 3D features, following the dense 3D features in DDet. So, in inference, SDet can simulate dense 3D features from regular (sparse) point cloud inputs without requiring dense inputs. We evaluate our method on the large-scale Waymo Open Dataset and the Waymo Domain Adaptation Dataset, showing its high performance and efficiency over the state of the arts. The code is available at https://github.com/stevewongv/Sparse2Dense.

## 1   Introduction

3D object detection is an important task for supporting autonomous vehicles to sense their surroundings. Previous works [1–4] design various neural network structures to improve the detection performance. Yet, it remains extremely challenging to detect small, distant, and incomplete objects, due to the point sparsity, object occlusion, and inaccurate laser reflection. These issues hinder further improvements in the precision and robustness of 3D object detection.

To improve the detection performance, some works attempt to leverage additional information, e.g., images [5–9], image segmentations [10, 11], and multi-frame information [12, 13]. By fusing the information with the input point cloud in

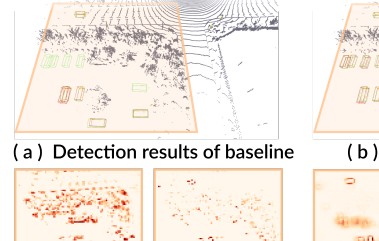

( a ) Detection results of baseline   ( b ) Our detection results

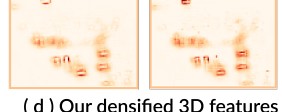

( c ) Sparse features visualization from baseline   ( d ) Our densified 3D features visualization from S2D

Figure 1: Our approach is able to produce dense and good-quality 3D features (d) from regular (raw) point clouds, enabling better detection of small, distant, and incomplete objects (b) vs. [1] (a,c). Red boxes are detection results and green boxes are the ground truths.

---

*Corresponding author (huxiaowei@pjlab.org.cn)

36th Conference on Neural Information Processing Systems (NeurIPS 2022).

physical or latent space, the 3D detector
can obtain enhanced features for small and distant objects to improve the 3D detection performance. However, the above works require additional data pre-processing and fusion in inference, thereby unavoidably increasing the computational burden and slowing down the overall detection efficiency in practice.

More recently, [14] associates incomplete perceptual features of objects with more complete features of the corresponding class-wise conceptual models via an incomplete-aware re-weighting map and a weighted MSE loss. However, this network still struggles to deal with sparse regions with limited points, due to the difficulty of generating good-quality features in these regions. Recently, [15] proposes to generate semantic points on the predicted object regions and then train modern detectors, leveraging both the generated and original points. However, as the generated points in sparse regions could be incomplete, the generation quality in these regions is still far from satisfactory. Also, it takes a long time to generate the semantic points in large scenes.

In this work, we present a new approach to address the point sparsity issue in 3D object detection. Specifically, we design the *Sparse2Dense framework* with two detectors: (i) the *Dense point 3D Detector (DDet)*, which is pre-trained with dense point clouds for 3D detection, and (ii) the *Sparse point 3D Detector (SDet)*, which is trained with regular (raw) point clouds as input. Very importantly, when we train SDet, *we use DDet to teach SDet to simulate densified 3D features*, so that it can learn to produce good-quality 3D features from regular point clouds to improve the detection performance. Unlike previous approaches, we design two effective modules to further help densify the sparse features from regular point clouds in latent space. Also, unlike previous multi-modal approaches [9, 10], which require extra information, like image segmentation and images, in both training and inference, our approach needs dense point data *only in training but not in inference*.

Our framework is trained in two stages. First, we prepare dense point clouds by fusing multi-frame point clouds for training DDet. Then, we transfer the densified 3D features derived from DDet for embedding into the SDet features when training SDet, such that it can learn to generate densified 3D features, even from regular normal point clouds. To facilitate SDet to simulate dense 3D features, we design the lightweight and effective *(S2D) module* to densify sparse 3D features in latent space. Also, to further enhance the feature learning, we formulate the *point cloud reconstruction (PCR) module* to learn to reconstruct voxel-level point cloud as an auxiliary task. Like DDet, we need this PCR module only in training but not in inference.

Furthermore, our framework is generic and compatible with various 3D detectors for boosting their performance. We adopted our framework to work with three different recent works [16, 17, 1] on the large-scale Waymo [18] Open and Waymo Domain Adaptation Datasets, showing that the detection performance of *all* three methods are improved by our approach. Particularly, the experimental results show that our approach outperforms the state-of-the-art 3D detectors on both datasets, demonstrating the effectiveness and versatility of our approach.

Below, we summarize the major contributions of this work.

(i) We design a new approach to address the point sparsity issue in 3D detection and formulate the Sparse2Dense framework to transfer dense point knowledge from the *Dense point 3D Detector* (DDet) to the *Sparse point 3D Detector* (SDet).

(ii) We design the lightweight plug-in S2D module to learn dense 3D features in the latent space and the point cloud reconstruction (PCR) module to regularize the feature learning.

(iii) We evaluate our approach on the large-scale benchmark datasets, Waymo [18] open and Waymo domain adaptation, demonstrating its superior performance over the state of the arts.

## 2   Related Work

**3D Object Detection.**   Recent years have witnessed the rapid progress of 3D object detection, existing works [19–31] have gained remarkable achievements on 3D object detection. SECOND [16] employs sparse convolution [32, 33] and PointPillars [17] introduces a pillar representation to achieve a good trade-off between speed and performance. Recently, CenterPoint [1] proposes a center-based anchor-free method to localize the object and VoTr [30] joins self-attention with sparse convolution to build a transformer-based 3D backbone. Besides, PV-RCNN [4] fuses deep features by RoI-grid pooling from both point and voxel features, and LiDAR R-CNN [26] presents a PointNet-based

second-stage refinement to address size ambiguity. In this work, we adopt our approach to three recent methods as representatives, i.e., [16, 17, 1], showing that our approach can effectively enhance the performance of *all* three methods, demonstrating the compatibility of our framework.

**Sparse/Dense Domain Transformation for 3D Point Cloud.** Raw point clouds are typically sparse and incomplete in practice, thereby limiting the performance of many downstream tasks. To address this issue, several approaches, including point cloud upsampling [34–36] and completion [37–40], have been proposed to densify point cloud and complete the objects to improve the 3D segmentation [41, 42] and detection [43, 14, 15, 31, 44] performance.

Here, we review some works on sparse/dense domain transformation on 3D object detection. [45] first propose a multi-frame to single-frame distillation framework, which only uses five adjacent frames to generate the dense features as the guidance, thus limiting the performance for distant objects. [43, 14] present a self-contained method to first extract complete cars at similar viewpoints and high-density regions across the dataset, then merge these augmented cars at the associated ground-truth locations for conceptual feature extraction. Later, [15] introduces semantic point generation to address the missing point issue. Recently, [44] presents a two-stage framework: first predict 3D proposals then complete the points by another module, which employs an attention-based GNN to refine the detection results with completed objects. The above works [43, 14, 15, 44] conduct various operations in the point cloud explicitly, *e.g.*, object extraction and matching [43, 14], and point generation [15, 44]; however, the explicit point cloud operations lead to two issues. First, it is challenging to conduct the above operations in distant and occluded regions, due to the high sparsity of points, thus severely limiting their performance in these regions. Second, it typically takes a very long time to conduct these operations explicitly, especially for large scenes.

Beyond the prior works, we present an efficient sparse-to-dense approach to learn to densify 3D features in the latent space, instead of explicitly generating points in the point cloud. Notably, our approach needs dense point clouds *only* in training. In inference, it takes only a regular (sparse) point cloud as input for 3D detection. Quantitative experiments demonstrate that our approach outperforms existing ones in terms of both accuracy and efficiency.

## 3 Methodology

### 3.1 Overall Framework Design

Figure 2 gives an overview of our Sparse2Dense framework, which consists of the *Dense point 3D Detector* (DDet) on top and the *Sparse point 3D Detector* (SDet) on bottom. Overall, DDet takes a dense point cloud as input, whereas SDet takes a regular (sparse) point cloud as input. Our idea is to transfer dense point knowledge from DDet to SDet and encourage SDet to learn to generate dense 3D features, even with sparse point clouds as inputs, to boost its 3D detection performance. The workflow of our framework design is summarized as follows:

(i) **Dense object generation:** we prepare dense point clouds for training DDet using raw multi-frame point cloud sequences. Particularly, we design a dense object generation procedure by building voxel grids and filling the voxels with object points (Section 3.2). Then, we replace the object regions in the sparse point cloud $P^S$ with the corresponding dense object points to obtain the dense point cloud $P^D$.

(ii) **From dense detector to sparse detector:** The training has two stages. First, as Figure 2 (a) shows, we train DDet with dense point cloud $P^D$ with a region proposal network (RPN) to extract region proposals and multiple heads to perform object classification and regression, following VoxelNet-based methods [1, 16] or Pillar-based method [17]. Second, as Figure 2 (b) shows, we initialize SDet with the weight of the pre-trained DDet, and train SDet with regular sparse point cloud $P^S$ as input. Meanwhile, we freeze the weights of DDet and adopt an MSE loss to reduce the feature difference ($F_a^D$ & $F_a^S$) between DDet and SDet.

(iii) **Dense feature generation by S2D module:** MSE loss itself is not sufficient to supervise SDet to effectively simulate dense 3D features like DDet. To complement the MSE loss, we further design the S2D module to learn dense 3D features of objects in latent space. In detail, we feed dense object point cloud $P_O^D$, which is the object region of the dense point cloud $P^D$, to DDet and then extract the dense object features $F_b^D$; After that, we further encourage feature $F_b^S$ enhanced by S2D to simulate to the dense object feature $F_b^D$.

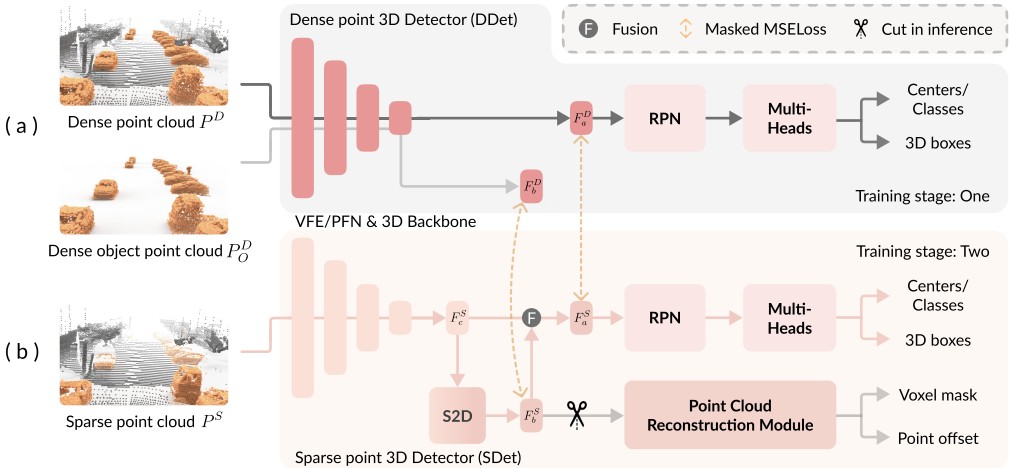

Figure 2: The overall framework of our proposed Sparse2Dense. Our framework contains two training stages: (a) in the first stage, we train the Dense point 3D Detector (DDet) by taking the dense point cloud as the input (dark arrows); and (b) in the second stage, we train the Sparse point 3D Detector (SDet) by using the dense features from DDet as the supervision signals (gray and pink arrows). In testing, we only need SDet for 3D object detection on the raw point cloud input (pink arrows), without the DDet and the point cloud reconstruction module.

(iv) **Feature enhancement by point cloud reconstruction (PCR) module:** further, we adopt the PCR module to promote the S2D module to simulate better dense 3D features; as an auxiliary task, PCR takes the feature from S2D module and predicts the voxel mask and point offset for reconstructing the voxel-level dense object point cloud $P_O^D$.

## 3.2 Dense Object Generation

To prepare dense point clouds to train the DDet network, we design an offline pre-processing pipeline to process raw point cloud sequences, each with around 198 frames (see Figure 3 (a)):

(i) First, for each annotated object, we fuse the points inside its bounding box from multiple frames and then filter out the outlier points by using the radius outlier removal algorithm from Open3D [46], as shown in Figure 3 (b).

(ii) To keep the LiDAR-scanned-line patterns and reduce the point number, we voxelize the fused object points and obtain a voxel grid (see Figure 3 (c)) of granularity $(0.1m, 0.1m, 0.15m)$, which is the same size for point cloud voxelization. Specifically, we sort the frames in descending order of the object point number, as shown in Figure 3 (a). Then, we fill the voxel grid with object points starting from the beginning of the sorted frames until more than $95\%$ voxels have been filled (see Figure 3 (d)). Note that we stop filling a single voxel when the number of points in the voxel has reached five or this voxel has been filled by the previous frames to obtain enough points for training.

(iii) For the vehicle category, whose shape is often symmetric, we flip and copy the denser side of each object about its axial plane to further improve its density (see Figure 3 (e)).

## 3.3 Dense Feature Distillation with the S2D Module

To transfer dense point knowledge from DDet to SDet, a straightforward solution is to pair up associated features in DDet and SDet and minimize the distance between each pair of features with an MSE loss. However, MSE loss itself is struggling to achieve satisfying feature transfer, as the inputs of DDet and SDet differ a lot from each other, especially for objects far from the LiDAR sensor. Also, the backbone structure [16, 1] built up with VoxelNet consists of SPConv, which processes only non-empty voxels. Hence, it cannot generate features on empty voxels.

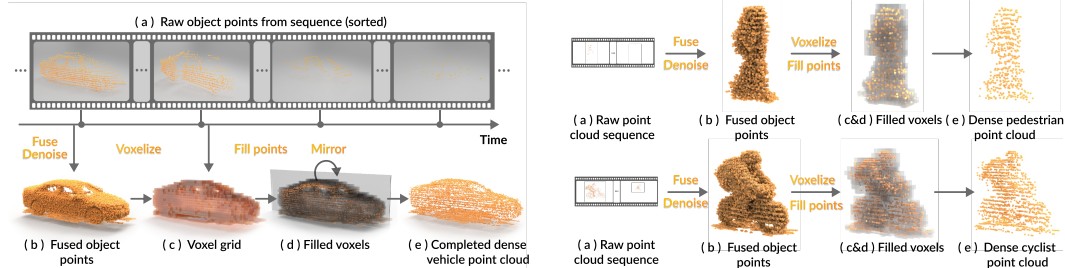

Figure 3: Dense object generation pipeline. Note that we use the annotated 3D bounding box to extract points from multiple frames and then fuse the points together with the help of a voxel grid.

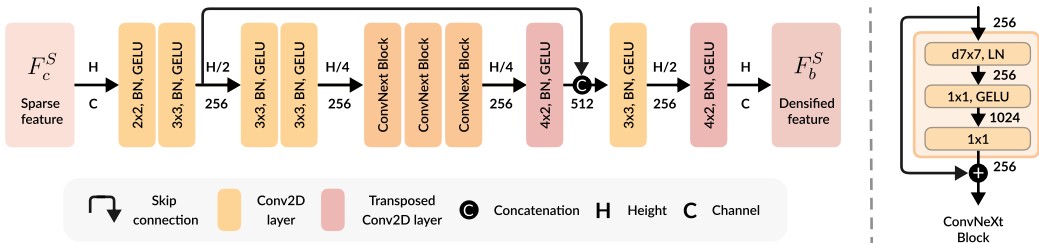

Figure 4: Left: the architecture of the S2D module. Right: ConvNeXt block [47].

To better densify the 3D features, we formulate the S2D module that takes SDet's backbone feature as input and learns to output denser features for 3D object detection. Figure 4 shows its architecture, in which we first project sparse 3D feature to obtain BEV feature $F_c^S$, so that we can employ efficient 2D convolution operations in the BEV space. Then, we down-sample the feature maps to $1/4$ size of the input sparse features by using convolution layers with stride 2. Inspired by the efficient design of convolution block in [47], we embed three ConvNeXt [47] residual blocks to aggregate the object information. Each block contains a $7 \times 7$ depth-wise convolution, followed by a layer normalization, a $1 \times 1$ conv with a GELU, and a $1 \times 1$ conv. As shown on the right of Figure 4, the first $1 \times 1$ conv increases the number of feature channels from 256 to 1024 and the second $1 \times 1$ conv reduces the channel number back to 256. Next, we upsample the features via a 2D transposed conv and concatenate the result with the previous features. After that, we feed the concatenated feature to a $3 \times 3$ conv layer and upsample the features to obtain the final densified feature $F_b^S$. Note that each conv, except convs in the ConvNeXt blocks, is followed by a batch normalization and a GELU non-linear operation. With the densified feature $F_b^S$, we fuse $F_b^S$ and $F_c^S$ by feeding each of them into a $1 \times 1$ conv layer and add them together as the final output feature $F_a^S$, as shown in Figure 2.

To train the S2D module, we consider two kinds of supervision. The first one is on the high-level features in the DDet network, where $F_a^D$ and $F_b^D$ are the features obtained on the dense point cloud with and without background information, respectively. We minimize the feature difference between $F_a^S$ and $F_a^D$ as well as between $F_b^D$ and $F_b^S$. The second supervision comes from the point reconstruction module to be presented in the next subsection.

## 3.4 Point Cloud Reconstruction Module

To encourage the S2D module to produce good-quality dense 3D features, we further design the point cloud reconstruction (PCR) module with an auxiliary task. In short, the PCR module reconstructs a voxel-level dense object point cloud $P_O^D$ from feature $F_b^S$, *i.e.*, the output of the S2D module. Yet, it is extremely challenging to directly reconstruct large-scale dense object points [15]. Thus, we propose a voxel-level reconstruction scheme to predict only the average of the input points in each non-empty voxel. Specifically, we decouple this task into two sub-tasks: first predict a soft voxel-occupancy mask indicating the probability that the voxel is non-empty; further predict point offset $P_{offset}$ for each non-empty voxel, *i.e.*, an offset from voxel center $V_c$ to the averaged input points of this voxel.

Figure 5 shows the architecture of the PCR module. The densified feature $F_b^S$ is first projected back to 3D view and fed to two 3D convolution layers and one transposed 3D convolution to upscale to $1/4$ scale of the original input. Then, we predict the voxel-occupancy mask $V_{mask}$ by using one $1 \times 1$ 3D convolution with sigmoid function and predict point offset $P_{offset}$ by using one $1 \times 1$ 3D convolution. Next, we repeat the same steps to further predict voxel mask $V_{mask}$ and point offset $P_{offset}$ at $1/2$ scale. Then, the reconstructed point $P_c$ is predicted as

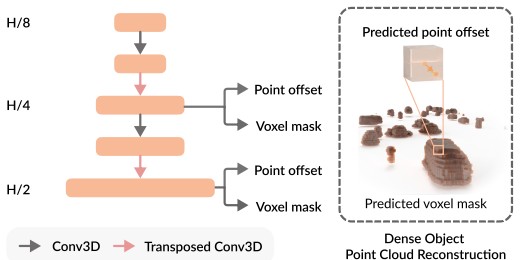

Figure 5: The architecture of the point cloud reconstruction (PCR) module.

$$P_c = (P_{offset} + V_c) \times V_{mask} , \tag{1}$$

and $P_c$ is optimized to reconstruct the voxelized dense object point cloud $P_O^D$.

## 3.5 Training Loss

Our Sparse2Dense framework is trained in two stages. First, we train DDet with the following loss:

$$\mathcal{L}_{\text{DDet}} = \mathcal{L}_{\text{reg}} + \mathcal{L}_{\text{hm/cls}} , \tag{2}$$

where we adopt an $L_1$ Loss as the regression loss $\mathcal{L}_{\text{reg}}$ followed [16, 1, 17]; heatmap loss $\mathcal{L}_{\text{hm}}$ is a variant of the focal loss [48] for center-based methods [1, 17]; and classification loss $\mathcal{L}_{\text{cls}}$ is the focal loss for anchor-based method [16]. Here, $\mathcal{L}_{\text{hm/cls}}$ means we use $\mathcal{L}_{\text{hm}}$ or $\mathcal{L}_{\text{cls}}$, depending which method that we adopt our framework to. Second, we train SDet with the following overall loss function:

$$\mathcal{L}_{\text{SDet}} = \mathcal{L}_{\text{reg}} + \mathcal{L}_{\text{hm/cls}} + \mathcal{L}_{\text{S2D}} + \mathcal{L}_{\text{mask}} + \mathcal{L}_{\text{offset}} + \mathcal{L}_{\text{hm\_dis}}, \tag{3}$$

where $\mathcal{L}_{\text{S2D}}$ is for optimizing S2D; $\mathcal{L}_{\text{mask}}$ and $\mathcal{L}_{\text{offset}}$ are for training PCR; and $\mathcal{L}_{\text{hm\_dis}}$ is the distillation loss like $\mathcal{L}_{\text{hm}}$, but its input is the predicted heat maps of SDet and DDet. In detail, $\mathcal{L}_{\text{S2D}}$ helps to optimize SDet to learn to densify 3D features based on the associated features in DDet and it is an MSE Loss with the masks to indicate the empty and non-empty elements in the feature maps:

$$\mathcal{L}_{\text{S2D}} = \beta \frac{1}{|N|} \sum_i^N (F_{a\,i}^S - F_{a\,i}^D)^2 + \gamma \frac{1}{|\widetilde{N}|} \sum_i^{\widetilde{N}} (F_{a\,i}^S - F_{a\,i}^D)^2$$
$$+ \beta \frac{1}{|M|} \sum_i^M (F_{b\,i}^S - F_{b\,i}^D)^2 + \gamma \frac{1}{|\widetilde{M}|} \sum_i^{\widetilde{M}} (F_{b\,i}^S - F_{b\,i}^D)^2, \tag{4}$$

where $N$ and $\widetilde{N}$ are numbers of non-zero and zero values, respectively, on $F_a^D$, while $M$ and $\widetilde{M}$ are numbers of non-zero and zero values, respectively, on $F_b^D$. Also, we empirically set $\beta$=10 and $\gamma$=20 to balance the loss weight on non-empty and empty features. $\mathcal{L}_{\text{mask}}$ and $\mathcal{L}_{\text{offset}}$ are for training PCR:

$$\mathcal{L}_{\text{mask}} = \sum_j \left( -\frac{N_b}{N_f} y_j \log(p_j) - (1 - y_j) \log(1 - p_j) \right) , \tag{5}$$

$$\text{and} \ \ \mathcal{L}_{\text{offset}} = \frac{1}{|N_f|} \sum_i^{N_f} |(P_{offset_i} + V_{c_i}) - P_{gt_i}| , \tag{6}$$

where $N_b$ and $N_f$ are numbers of background and foreground voxels, respectively; $p_j$ and $y_j$ are the prediction and ground-truth values of the voxel mask; and $j$ indexes the voxels in $V_{mask}$. Note also that Eq. (3) includes $\mathcal{L}_{\text{hm\_dis}}$, only when adopting our method to the center-based methods [1, 17].

## 4 Experiments

### 4.1 Datasets and Evaluation Metrics

We employ the Waymo Open Dataset and the Waymo Domain Adaptation Dataset [18], which are under the Waymo Dataset License Agreement, to evaluate our framework. Waymo Open Dataset is

Table 1: Comparisons on the Waymo Open Dataset on 202 validation sequences with existing works. † means re-produced by [28, 4]. ⋆ means re-produced by [14]. ‡ means re-produced by us. Note that our re-produced models and most of other re-produced models were trained on 20% data of Waymo Open Dataset following the strategy of [28, 4]. See more details of implementation in Sec. 4.2 and comparison in Sec. 4.3. Note that [30, 29] were trained only on the vehicle category, so they can largely focus on learning features for vehicles, [15] was trained on the vehicle and pedestrian categories, and others were trained on all three categories.

| Methods | Vehicle-L1 | | Pedestrian-L1 | | Cyclist-L1 | | Vehicle-L2 | | Pedestrian-L2 | | Cyclist-L2 | |
|---|---|---|---|---|---|---|---|---|---|---|---|---|
| | mAP | mAPH | MAP | mAPH | mAP | mAPH | mAP | mAPH | MAP | mAPH | mAP | mAPH |
| Part-A2-Net† [28] | 71.82 | 71.29 | 63.15 | 54.96 | 65.23 | 63.92 | 64.33 | 63.82 | 54.24 | 47.11 | 62.61 | 61.35 |
| VoTr-SSD [30] | 68.99 | 68.39 | - | - | - | - | 60.22 | 59.69 | - | - | - | - |
| VoTr-TSD [30] | 74.95 | 74.95 | - | - | - | - | 65.91 | 65.29 | - | - | - | - |
| Pyramid-PV [29] | 76.30 | 75.68 | - | - | - | - | 67.23 | 66.68 | - | - | - | - |
| **Densified:** | | | | | | | | | | | | |
| PV-RCNN† [4] | 74.06 | 73.38 | 62.66 | 52.68 | 63.32 | 60.72 | 64.99 | 64.38 | 53.80 | 45.14 | 60.72 | 59.18 |
| PV-RCNN + SPG [15] | 75.27 | - | 66.93 | - | - | - | 65.98 | - | 57.68 | - | - | - |
| SECOND⋆ [16] | 67.40 | 66.80 | 57.40 | 47.80 | 53.50 | 52.30 | 58.90 | 58.30 | 49.40 | 41.10 | 51.80 | 50.60 |
| SECOND+AGO-Net⋆ [14] | 69.20 | 68.70 | 59.30 | 48.70 | 55.30 | 54.20 | 60.60 | 60.10 | 51.80 | 42.40 | 53.50 | 52.50 |
| SECOND‡ [16] | 67.49 | 66.06 | 55.59 | 44.66 | 57.32 | 54.54 | 59.42 | 57.92 | 47.99 | 38.50 | 55.19 | 52.51 |
| **SECOND (ours)** | **71.94** | **70.47** | **58.78** | **48.29** | **59.24** | **56.76** | **63.49** | **62.17** | **51.12** | **41.92** | **57.03** | **54.64** |
| CenterPoint-Pillar‡ [17, 1] | 72.36 | 71.73 | 69.16 | 59.16 | 62.11 | 60.42 | 64.12 | 63.54 | 61.14 | 52.13 | 59.76 | 58.14 |
| **CenterPoint-Pillar(ours)** | **76.10** | **75.53** | **74.29** | **65.20** | **67.81** | **66.22** | **68.11** | **67.58** | **66.41** | **58.06** | **65.28** | **63.74** |
| CenterPoint‡ [1] | 73.70 | 72.96 | 74.73 | 69.07 | 68.85 | 67.73 | 65.52 | 65.01 | 66.30 | 61.09 | 66.32 | 65.24 |
| **CenterPoint(ours)** | **76.09** | **75.52** | **78.22** | **72.50** | **71.95** | **70.83** | **68.21** | **67.68** | **70.07** | **64.72** | **69.31** | **68.23** |

the largest and most informative 3D object detection dataset, which includes 360° LiDAR point cloud and annotated 3D bounding boxes. The training set contains 798 sequences with around 158K LiDAR frames and the validation set includes 202 sequences with around 40k LiDAR frames. The dataset is captured across California and Arizona. The labeled object categories include vehicle, pedestrian, and cyclist. All the objects in sequences are named with a unique ID that can be used to generate the dense object point cloud in our method. Also, we perform unsupervised domain adaptation on the Waymo Domain Adaptation dataset without re-training our model to show the generalization capability of our method. The Waymo Domain Adaptation dataset contains 20 sequences with 3933 frames for evaluation. This dataset is captured in Kirkland and most frames are captured on rainy day [15], which means the point cloud is sparser and more incomplete than the point cloud in Waymo Open Dataset. The labeled object categories include vehicle and pedestrian in the Waymo Domain Adaptation dataset. Following prior works [1, 4], we adopt Average Precision weighted by Heading (APH) and Average Precision (AP) as evaluation metrics.

## 4.2 Implementation Details

In the first training stage, following [1], we train DDet from scratch using Adam with a learning rate of 0.003 and a one-cycle learning rate policy with a dividing factor of 0.1 and a percentage of the cycle of 0.3. We set the detect range as $[-75.2m, 75.2m]$ for the $X, Y$ axes and set $[-2m, 4m]$ for the $Z$ axis, and the size of each voxel grid as $(0.1m, 0.1m, 0.15m)$. We apply global rotation around the Z-axis, random flipping, global scaling, and global translating as the data augmentation. We train the DDet on four Nvidia RTX 3090 GPUs with a batch size of four per GPU for 30 epochs.

In the second training stage, we adopt DDet's weights to initialize SDet, then optimize SDet by adopting the same hyperparameters as the first stage with DDet frozen. Following [4, 14, 30], we adopt 20% subset of the Waymo Open Dataset to train our models. Note that we adopt the second stage for CenterPoint-based baselines [1, 17] by following [1].

## 4.3 Comparison with State-of-the-art Methods on the Waymo Open Dataset

We compare our methods with multiple state-of-the-art 3D object detectors [28, 4, 30, 29, 15–17, 1] on three categories, *i.e.*, pedestrian, vehicle, and cyclist, and on two difficulty levels, *i.e.*, level 1 (L1) and level 2 (L2). Note that our method is a plug-and-play module and can be easily adopted to work with various deep-learning-based 3D object detectors. We obtain the results of the existing works by copying from their papers and GitHub [28, 4, 14, 30, 29, 15] or by re-producing their methods using the public code with recommended parameters [1, 16]. We cannot compare with [44], as the authors

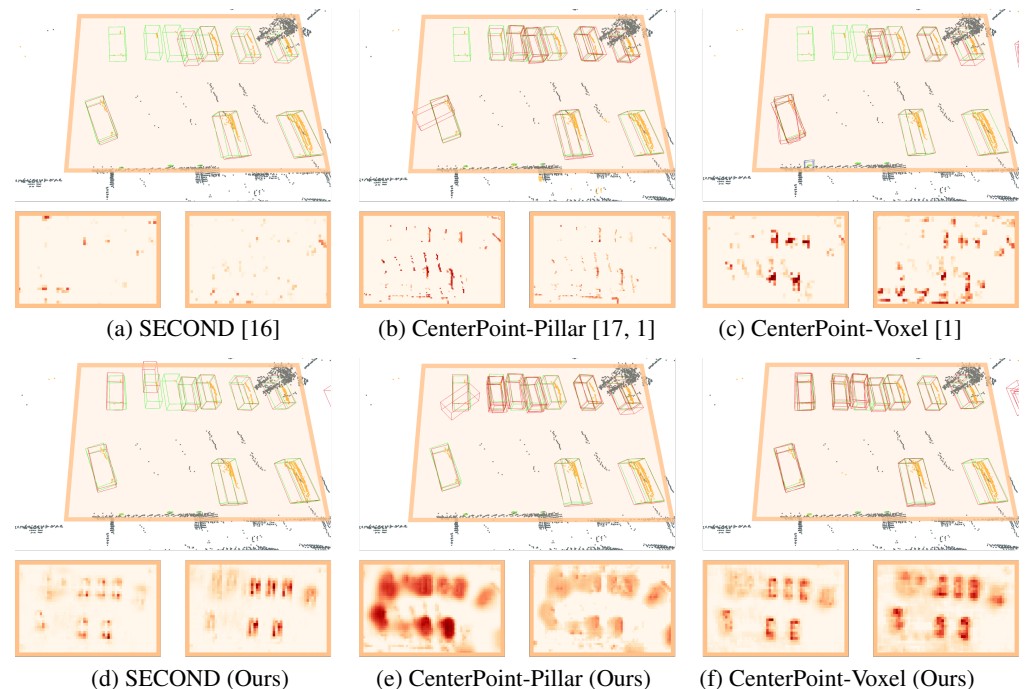

|          |          |          |
| :------: | :------: | :------: |
| (a) SECOND [16] | (b) CenterPoint-Pillar [17, 1] | (c) CenterPoint-Voxel [1] |
| (d) SECOND (Ours) | (e) CenterPoint-Pilllar (Ours) | (f) CenterPoint-Voxel (Ours) |

Figure 6: Visual comparison of 3D object detection results and 3D features produced by (a,b,c) baselines [16, 17, 1] and (d,e,f) our methods (our approach + corresponding baseline), where our approach successfully densifies the object features and help the baseline methods produce more accurate detection results than the three baselines. Note that red boxes show the detection results and green boxes show the ground truths. Orange boxes highlight the improvement brought by our approach.

did not report the performance of their method on the Waymo Open Dataset and did not release code. Among the existing works, [15, 14] are the state-of-the-art methods that adopt densified operations explicitly in point clouds and can be adopted to work with other methods.

Table 1 reports the comparison results, where our method clearly improves *all* three baseline methods (SECOND, PointPilllar-center, and CenterPoint) for *all* categories on *all* evaluation metrics. Notably, our approach achieves more performance gain in comparison than [14] when working with [16], and outperforms [15] when working with [17, 1], even though [15] is built upon a stronger model [4]. Also, level 2 (L2) contains more challenging samples than level 1 (L1), since the point cloud in L2 are much sparser. Despite that, our method *consistently* shows greater improvement on L2, demonstrating its effectiveness to deal with the sparse point clouds. Furthermore, Table 2 shows the performance gain for three different distant ranges on the Waymo validation set. Our method also achieves significant improvements over *all* three baseline methods for *all* distance ranges, including the long-range, showing again that our method can effectively learn to densify 3D features in challenging cases. See more detailed comparison results and the performance of our model trained on the full Waymo Open Dataset in Appendix A. We further provide the visual comparisons in Figure 6, where we can see that: (i) our method successfully detects more objects that contain a few points and compensates sparse features of these objects; (ii) our method generates more accurate 3D bounding boxes, which are consistent with the ground truth boxes; see the orange boxes in the first row; (iii) our method generates more dense and robust features in sparse or distant regions. Please see more visual comparisons in Appendix B.

## 4.4 Quantitative Comparison on the Waymo Domain Adaptation Dataset

We further evaluate our methods on the Waymo Domain Adaptation Dataset and compare it with the state-of-the-art methods. Table 3 shows that our method is able to consistently improve the performance of *all three methods* [16, 17, 1] for *all categories* and *all difficulty levels*. Note also

Table 2: Performance gain over baseline approaches on Waymo validation set (level-2) in different ranges. Evaluated on three categories.

| Methods | All Range | | Range [0, 30) | | Range [30, 50) | | Range [50, +inf) | |
|---|---|---|---|---|---|---|---|---|
| | mAP | mAPH | MAP | mAPH | mAP | mAPH | mAP | mAPH |
| SECOND [16] | 54.20 | 49.65 | 70.58 | 66.17 | 52.33 | 46.95 | 32.64 | 28.25 |
| **SECOND (ours)** | 57.21 **+3.01** | 52.91 **+3.26** | 72.94 **+2.36** | 68.91 **+2.74** | 55.30 **+2.97** | 50.28 **+3.33** | 36.32 **+3.68** | 31.96 **+3.71** |
| CenterPoint-Pillar [17, 1] | 61.67 | 57.94 | 74.18 | 70.55 | 61.63 | 57.88 | 42.91 | 38.83 |
| **CenterPoint-Pillar(ours)** | 66.60 **+4.93** | 63.13 **+5.19** | 77.90 **+3.72** | 74.60 **+4.05** | 66.72 **+5.09** | 63.32 **+5.44** | 49.42 **+6.51** | 45.41 **+6.58** |
| CenterPoint-Voxel [1] | 66.04 | 63.78 | 80.80 | 78.86 | 64.24 | 61.66 | 45.37 | 42.35 |
| **CenterPoint-Voxel(Ours)** | 69.19 **+3.15** | 66.88 **+3.10** | 82.72 **+1.92** | 80.77 **+1.91** | 67.60 **+3.36** | 64.96 **+3.30** | 49.45 **+4.08** | 46.28 **+3.93** |

Table 3: Comparisons on the Waymo Domain Adaptation Dataset on 20 validation sequences with existing works. [†] means re-produced by [15], [‡] means re-produced by us. Still, our re-produced models and most of other re-produced models were trained on 20% data of Waymo Open Dataset following the strategy of [28]; see more details in Sec. 4.2. [15] was trained on two categories (vehicle and pedestrian) while ours were trained on all the three categories.

| Methods | Vehicle-L1 | | Pedestrian-L1 | | Vehicle-L2 | | Pedestrian-L2 | |
|---|---|---|---|---|---|---|---|---|
| | mAP | mAPH | mAP | mAPH | mAP | mAPH | mAP | mAPH |
| SPG[†] [15] | 58.31 | - | 30.82 | - | 48.70 | - | 22.05 | - |
| SECOND[‡] [16] | 51.56 | 49.55 | 13.96 | 12.14 | 42.90 | 41.22 | 9.83 | 8.54 |
| **SECOND(ours)** | **55.49** | **53.96** | **17.45** | **15.25** | **46.25** | **44.95** | **12.23** | **10.68** |
| CenterPoint-Pillar[‡] [17, 1] | 54.15 | 53.26 | 12.50 | 10.36 | 45.33 | 44.57 | 8.80 | 7.29 |
| **CenterPoint-Pillar(ours)** | **59.18** | **58.52** | **18.95** | **16.26** | **50.12** | **49.55** | **13.31** | **11.42** |
| CenterPoint-Voxel[‡] [1] | 57.54 | 56.99 | 30.21 | 28.30 | 48.36 | 47.88 | 21.16 | 19.82 |
| **CenterPoint-Voxel(ours)** | **60.54** | **59.87** | **37.15** | **35.21** | **51.01** | **50.43** | **26.03** | **24.66** |

that the most recent state-of-the-art method SPG [15] is trained only on the vehicle and pedestrian categories, yet our method can achieve better performance even when trained on three categories.

## 4.5 Ablation Study

We conduct experiments to evaluate the key components in our Sparse2Dense framework. Here, we adopt our approach to work with CenterPoint (one stage version) [1]. Then, we conduct the feature distillation ("+ Distillation") to distill the 3D features from $F_a^D$ in DDet to $F_a^S$ in SDet, and adopt heat map distillation $\mathcal{L}_{hm\_dis}$ between the heat maps of SDet and DDet, as discussed in Sec.3.5. Next, we further add the S2D module ("+ S2D") in SDet to enable the feature distillation between the $F_b^D$ in DDet and $F_b^S$ in SDet. In addition, we construct our full pipeline by further adding the point cloud reconstruction module ("+ PCR"). Also, we conduct an additional experiment ("- Distillation") by ablating the feature loss $\mathcal{L}_{S2D}$ and heat map distillation loss $\mathcal{L}_{hm\_dis}$ from our full pipeline.

The results are shown in Table 4. First, feature distillation ("+ Distillation") helps to moderately improve the performance of both categories. By adopting our S2D module ("+ S2D"), we can largely improve the quality of the densified features, boosting the performance of 3D object detection by around 2% on the vehicle and 1% on the cyclist, compared with "+ Distillation". Next, our point cloud reconstruction module ("+ PCR") further enhances the performance on both categories and metrics consistently by providing additional supervision to regularize the feature learning. Finally, even without distillation ("- Distillation"), our approach can still improve the baseline performance by more than 2.5% on the vehicle, demonstrating the effectiveness of our S2D and PCR modules.

## 4.6 Latency Analysis for S2D module

In our framework, both DDet and PCR are used only in the training stages. In inference, we only add the lightweight S2D module into the basic 3D object detection framework. To evaluate the efficiency of the S2D module, we employ the Waymo Open Dataset validation set and report the average processing time with and without the S2D module in inference. Table 5 reports the results, showing that S2D only brings around extra 10 ms latency to detectors, thus demonstrating our approach's high efficiency. Also, we show the latency of SPG [15] reported in their paper, $i.e.$, 16.9 ms, which is evaluated on the KITTI dataset with a much smaller number of points and objects than the dataset we employed, $i.e.$, Waymo. The latency analysis results manifest the superior efficiency of our S2D in comparison to the state-of-the-art approach [15].

Table 4: Ablation studies on the Waymo Open Dataset validation set.

| Methods | Vehicle-L2 | | Pedestrian-L2 | | Cyclist-L2 | |
|---|---|---|---|---|---|---|
| | mAP | mAPH | mAP | mAPH | mAP | mAPH |
| Baseline | 63.03 | 62.53 | 63.72 | 58.03 | 65.03 | 63.90 |
| + Distillation | 63.84 | 63.32 | 67.04 | 61.21 | 67.59 | 66.44 |
| + S2D | 65.75 | 65.22 | **67.62** | **61.65** | 68.50 | 67.34 |
| + PCR | **66.12** | **65.58** | 67.47 | 61.59 | **68.69** | **67.54** |
| - Distillation | 65.61 | 65.08 | 64.75 | 58.80 | 65.79 | 64.62 |

Table 5: Latency analysis on our S2D module. We evaluate each model with the batch size of 1. The latency is averaged over the Waymo validation set. As a reference, we include SPG [15] (evaluated on KITTI). Our method needs only ~10 ms vs. 16.9 ms by SPG.

| Detectors | CenterPoint-Pillar [17, 1] | CenterPoint-Pillar+S2D | CenterPoint-Voxel [1] | CenterPoint-Voxel+S2D |
|---|---|---|---|---|
| Inference time (ms) | 42.7 | 53.1 (+10.4) | 53.0 | 62.8 (+9.8) |

| Detectors | PV-RCNN [4] | PV-RCNN+SPG [15] |
|---|---|---|
| Inference time (ms) | 140.0 | 156.9 (+16.9) |

## 5  Discussion and Conclusion

This paper presents the novel Sparse2Dense framework that learns to densify 3D features to boost 3D object detection performance. Our key idea is to learn to transfer dense point knowledge from the trained dense point 3D detector (DDet) to the sparse point 3D detector (SDet), such that SDet can learn to densify 3D features in the latent space. With the trained SDet, we only need the core component of SDet to detect 3D objects in regular point clouds, so we can enhance the detection accuracy without degrading the speed. Further, to enhance the transfer of dense point knowledge, we design the S2D module and the point cloud reconstruction module in SDet to enhance the sparse features. Last, we adopt our framework to various 3D detectors, showing that their performance can all be improved consistently on multiple benchmark datasets. In the future, we will apply our framework to more point cloud applications that require dense features, such as 3D segmentation and object tracking, to boost their performance while maintaining high computational efficiency.

**Limitations.** First, objects far from the LiDAR sensor in the point cloud sequence contain only a few points. It is still difficult to generate dense features for these objects with our DDet. Second, the training time of our framework is longer than traditional training, as we need multi-stage training to pre-train DDet and then transfer knowledge from the pre-trained DDet to SDet. Third, our models trained on Waymo Open Dataset need inputs containing specific point features like intensity and elongation, which limits our models evaluating on different datasets, like KITTI. We will explore removing the specific point features to make the model more general in future works.

**Societal Impacts.** Our proposed framework can provide better 3D object detection performance for autonomous vehicles. However, like most existing 3D detectors, it may produce errors in some edge cases, due to the limited data, so further research is still needed to improve its robustness.

**Acknowledgements.** Thank all the co-authors, reviewers, and ACs for their remarkable efforts. This work was supported by the project #MMT-p2-21 of the Shun Hing Institute of Advanced Engineering, The Chinese University of Hong Kong, and the Shanghai Committee of Science and Technology (Grant No.21DZ1100100).

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
