# Sparse2Dense: Learning to Densify 3D Features for 3D Object Detection

**Tianyu Wang**[1,2,3]**, Xiaowei Hu**[3,*]**, Zhengzhe Liu**[1]**, Chi-Wing Fu**[1,2]

[1] The Chinese University of Hong Kong
[2] The Shun Hing Institute of Advanced Engineering
[3] Shanghai AI Laboratory
{wangty,zzliu,cwfu}@cse.cuhk.edu.hk, huxiaowei@pjlab.org.cn

In this Appendices, we provide the following two parts:

- **Appendix A** provides detailed performance gain information on the Waymo validation set (level 2), showing that our framework can improve the detection performance for different distance ranges over three categories, especially distant objects that only contain a few points. In this part, we also provide the performance comparison when models were trained on the full Waymo Train set, which demonstrates the effectiveness of our framework.

- **Appendix B** presents the feature maps in point clouds that have been densified by our approach, showing that our approach is able to effectively densify and enrich the object features, helping the detector to handle more distant and occluded objects.

---

*Corresponding author (huxiaowei@pjlab.org.cn)

36th Conference on Neural Information Processing Systems (NeurIPS 2022).

# Appendix A  Additional Experiments

## A.1  Detailed Performance Gain on the Waymo Validation Set (Level 2)

In this subsection, we present detailed performance comparisons over three baselines [1–3] on the Waymo validation set (level-2) for different distance ranges and three categories. We show the percentage improvement brought forth by our approach on various cases (baselines and distance ranges), demonstrating that our approach helps various methods to improve their performance.

Table A1: Performance gain over three baselines on Waymo validation (level-2) for different distance ranges. Evaluated on **vehicle**. Our approach largely improves the performance on distant objects.

| Methods | All Range | | Range [0, 30) | | Range [30, 50) | | Range [50, +inf) | |
|---|---|---|---|---|---|---|---|---|
| | mAP | mAPH | MAP | mAPH | mAP | mAPH | mAP | mAPH |
| SECOND [1] | 59.42 | 57.92 | 86.39 | 84.92 | 58.98 | 56.92 | 30.27 | 28.81 |
| **SECOND (ours)** | 63.49 **+4.07** | 62.17 **+4.25** | 88.72 **+2.33** | 87.50 **+2.58** | 63.39 **+4.41** | 61.72 **+4.80** | 35.63 **+5.36** | 34.27 **+5.46** |
| CenterPoint-Pillar [2, 3] | 64.12 | 63.54 | 88.35 | 87.77 | 63.88 | 63.29 | 36.29 | 35.62 |
| **CenterPoint-Pillar(ours)** | 68.11 **+3.99** | 67.58 **+4.04** | 90.01 **+1.74** | 89.51 **+1.55** | 68.28 **+4.15** | 67.72 **+4.2** | 41.92 **+6.27** | 41.26 **+6.30** |
| CenterPoint-Voxel [3] | 65.52 | 65.01 | 89.47 | 88.97 | 66.02 | 65.47 | 37.46 | 36.89 |
| **CenterPoint-Voxel(Ours)** | 68.21 **+2.69** | 67.68 **+2.67** | 90.23 **+0.76** | 89.76 **+0.79** | 68.70 **+2.68** | 68.11 **+2.64** | 41.81 **+4.35** | 41.13 **+4.24** |

Table A2: Performance gain over three baselines on Waymo validation (level-2) for different distance ranges. Evaluated on **pedestrian**. Our approach largely improves the performance on distant objects.

| Methods | All Range | | Range [0, 30) | | Range [30, 50) | | Range [50, +inf) | |
|---|---|---|---|---|---|---|---|---|
| | mAP | mAPH | MAP | mAPH | mAP | mAPH | mAP | mAPH |
| SECOND [1] | 47.99 | 38.50 | 56.58 | 47.09 | 48.00 | 37.26 | 32.53 | 24.03 |
| **SECOND (ours)** | 51.12 **+3.13** | 41.92 **+3.42** | 59.32 **+3.36** | 50.45 **+2.77** | 51.17 **+3.42** | 40.68 **+3.19** | 36.39 **+4.30** | 27.46 **+3.99** |
| CenterPoint-Pillar [2, 3] | 61.14 | 52.13 | 64.92 | 56.29 | 65.13 | 55.77 | 49.29 | 39.83 |
| **CenterPoint-Pillar(ours)** | 66.41 **+5.27** | 58.06 **+5.93** | 70.36 **+5.44** | 62.54 **+6.25** | 69.25 **+4.12** | 67.72 **+5.02** | 41.92 **+6.13** | 41.26 **+6.17** |
| CenterPoint-Voxel [3] | 66.30 | 61.09 | 74.77 | 70.48 | 66.68 | 60.69 | 50.64 | 43.49 |
| **CenterPoint-Voxel(Ours)** | 70.07 **+3.77** | 64.72 **+3.63** | 77.98 **+3.21** | 73.62 **+3.14** | 70.26 **+3.58** | 64.16 **+3.47** | 55.31 **+4.67** | 47.80 **+4.31** |

Table A3: Performance gain over three baselines on Waymo validation (level-2) for different distance ranges. Evaluated on **cyclist**. Our approach largely improves the performance on distant objects.

| Methods | All Range | | Range [0, 30) | | Range [30, 50) | | Range [50, +inf) | |
|---|---|---|---|---|---|---|---|---|
| | mAP | mAPH | MAP | mAPH | mAP | mAPH | mAP | mAPH |
| SECOND [1] | 55.19 | 52.51 | 68.77 | 66.50 | 50.00 | 46.69 | 35.11 | 31.90 |
| **SECOND (ours)** | 57.03 **+1.84** | 54.64 **+2.13** | 70.78 **+2.01** | 68.79 **+2.29** | 51.36 **+1.36** | 48.43 **+1.74** | 36.93 **+1.82** | 34.13 **+2.23** |
| CenterPoint-Pillar [2, 3] | 59.76 | 58.14 | 69.26 | 67.58 | 55.87 | 54.59 | 43.16 | 41.03 |
| **CenterPoint-Pillar(ours)** | 65.28 **+5.52** | 63.74 **+5.60** | 73.34 **+4.08** | 71.74 **+4.16** | 62.62 **+6.75** | 61.46 **+6.87** | 50.93 **+7.77** | 48.96 **+7.93** |
| CenterPoint-Voxel [3] | 66.32 | 65.24 | 78.16 | 77.13 | 60.02 | 58.82 | 48.01 | 46.66 |
| **CenterPoint-Voxel(Ours)** | 69.34 **+2.99** | 68.23 **+2.99** | 79.93 **+1.77** | 78.92 **+1.79** | 63.85 **+3.83** | 62.62 **+3.80** | 51.22 **+3.21** | 49.90 **+3.24** |

## A.2  Performance Comparison When Models Trained on Full Waymo *Train* set (Level 2)

We follow [3] to train our model with full ***train*** set of Waymo Open Dataset and test the trained model on Waymo ***val*** set and ***test*** set. Table A4 shows that our method surpasses the baseline method. Note that we only train our model in a short training schedule (1x means training 12 epochs for the first stage) due to the limited computational resources, but we still achieve better results.

Table A4: Performance Comparison on the Waymo ***val*** set and ***test*** Set. (Level 2)

| Methods | Split | Schedule | Vehicle-mAP | Pedestrian-mAP | Cyclist-mAP |
|---|---|---|---|---|---|
| CenterPoint-Voxel | Val | 3x | 67.9 | 65.6 | 68.6 |
| **CenterPoint-Voxel(Ours)** | Val | 1x | **68.4** | **71.2** | **71.3** |
| CenterPoint-Voxel | Test | 3x | 71.9 | 67.0 | 68.2 |
| **CenterPoint-Voxel(Ours)** | Test | 1x | **72.6** | **72.1** | **70.3** |

# Appendix B    Additional Comparison Results with Feature Visualization

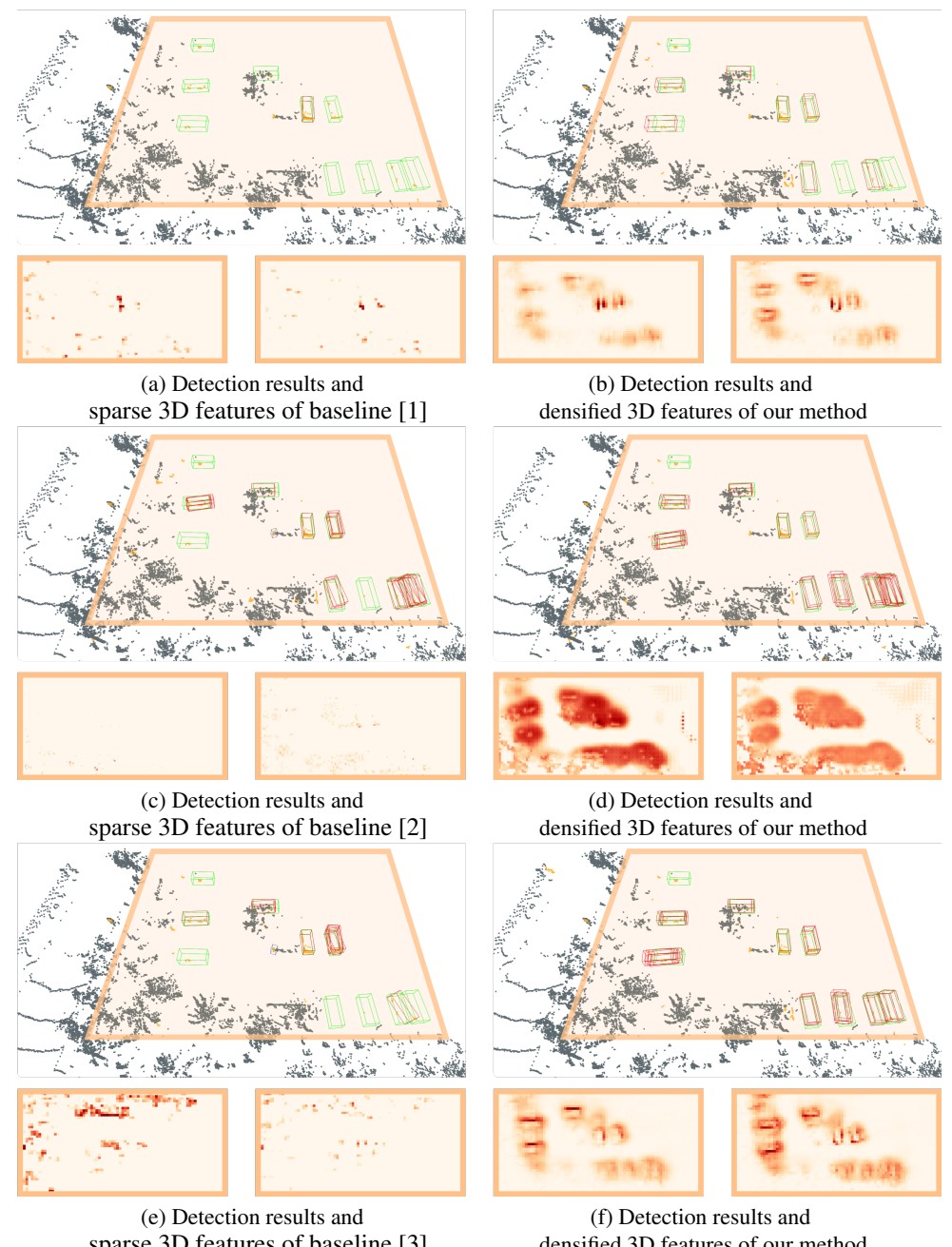

Figure A1: Visual comparison of 3D object detection results and 3D features produced by (a,c,e) baselines [1–3] and (b,d,f) our methods (our approach + corresponding baseline), where our approach successfully densifies the object features and helps the baseline methods produce more accurate detection results than three baselines. Note that red boxes show the detection results and green boxes show the ground truths. Orange boxes highlight the improvement brought by our approach.

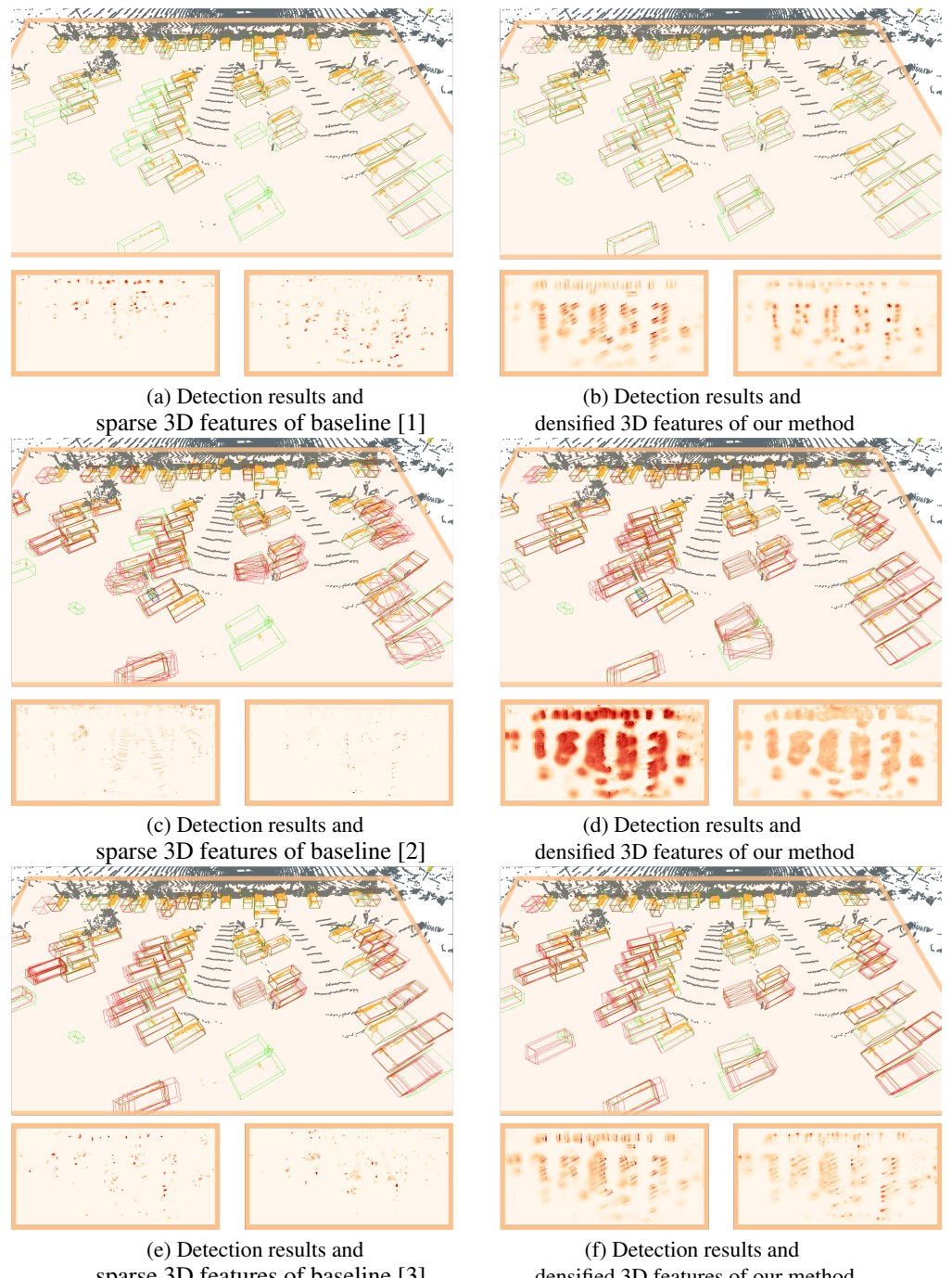

(a) Detection results and
sparse 3D features of baseline [1]

(b) Detection results and
densified 3D features of our method

(c) Detection results and
sparse 3D features of baseline [2]

(d) Detection results and
densified 3D features of our method

(e) Detection results and
sparse 3D features of baseline [3]

(f) Detection results and
densified 3D features of our method

Figure A2: Visual comparison of 3D object detection results and 3D features produced by (a,c,e)
baselines [1–3] and (b,d,f) our methods (our approach + corresponding baseline), where our approach
successfully densifies the object features and helps the baseline methods produce more accurate
detection results than three baselines. Note that red boxes show the detection results and green boxes
show the ground truths. Orange boxes highlight the improvement brought by our approach.

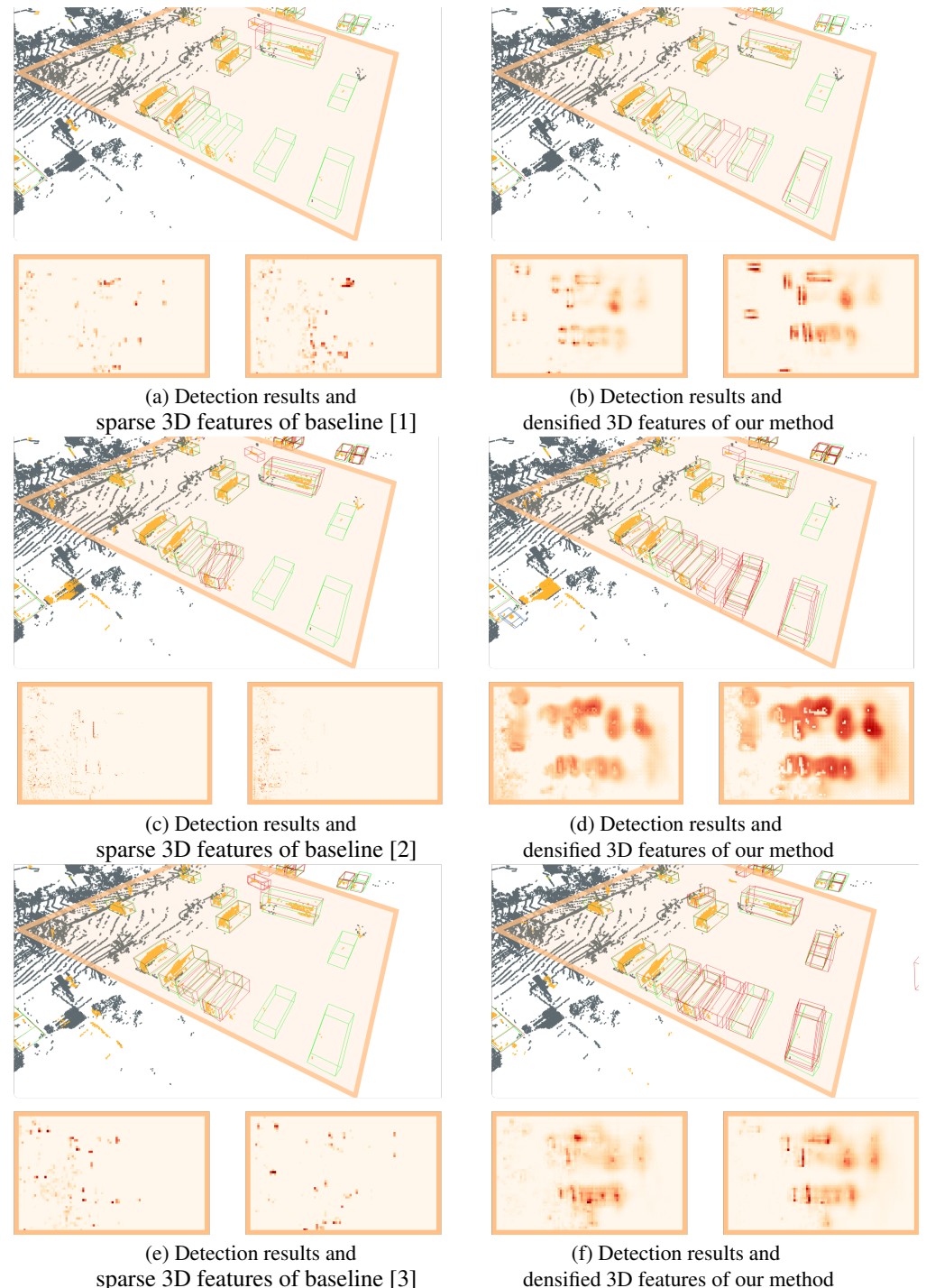

(a) Detection results and
sparse 3D features of baseline [1]

(b) Detection results and
densified 3D features of our method

(c) Detection results and
sparse 3D features of baseline [2]

(d) Detection results and
densified 3D features of our method

(e) Detection results and
sparse 3D features of baseline [3]

(f) Detection results and
densified 3D features of our method

Figure A3: Visual comparison of 3D object detection results and 3D features produced by (a,c,e) baselines [1–3] and (b,d,f) our methods (our approach + corresponding baseline), where our approach successfully densifies the object features and helps the baseline methods produce more accurate detection results than three baselines. Note that red boxes show the detection results and green boxes show the ground truths. Orange boxes highlight the improvement brought by our approach.