# OpenReview forum: "Sparse2Dense: Learning to Densify 3D Features for 3D Object Detection"
_NeurIPS.cc/2022/Conference — NeurIPS 2022 Accept_

### Official Review · Reviewer_vmwQ · 2022-07-06

**Rating:** 5
**Confidence:** 5
**Soundness:** 3 good
**Presentation:** 2 fair
**Contribution:** 3 good

**Summary:**

This paper presents a new framework to efficiently boost 3D detection performance by learning to densify point clouds in latent space. And experiments on Waymo Open Dataset show promising results.

**Questions:**

see part ‘Strengths And Weaknesses’.

**Limitations:**

see part ‘Strengths And Weaknesses’.

**Strengths And Weaknesses:**

1. By learning the feature representation of the point cloud in the dense point cloud space and then transferring it to the sparse point cloud space, this idea is relatively novel and feasible.

2. Although the transition from dense to sparse is a good idea, there is still a large gap from dense to sparse. Especially when there is severe occlusion, this gap can be particularly large. And this case is a very common problem in the large practical application of point cloud target detection. So what about the performance in this bad case?

3. The process of learning the feature representation of the point cloud in the dense point cloud space is actually an upsampling of the point cloud, and the method of voxelization is actually not an optimal way. Have you tried any other way? And have you compared the effects of different upsampling methods?

4. The method in this paper is mainly used for experiments on the waymo dataset, so how about the performance on other datasets? Such as KITTI, nuScenes.

5. The method in this paper is only compared with a few methods in the experimental results. In fact, there are many excellent works in the point cloud target detection task, but the comparison with them is missing.

---

> ### Author Response · Authors · 2022-08-02
> **Response to Reviewer vmwQ**
>
> For S&W 2:
> * We think that Figure 1 presents a good example. The orange region in the figure is the distant region and each object contains a few points. Our method can still achieve an acceptable performance in this bad case.
>
> For S&W 3:
> * To our knowledge, the upsampling only generates points near the original points and it can not recover the missing structure. Some papers try to upsample the whole frame first, but it generates a lot of insignificant points (like background points), or just complete the objects for the proposals and then refine the proposals [43], which is very time consuming. This is why we choose to densify the feature in the latent space by formulating a light-weight module.
>
> For S&W 4:
> * Thanks for your suggestion! Actually, the Waymo Domain Adaptation dataset is much more challenging than the training dataset, which is the Waymo open dataset. It is because the data was captured in a different city in a rainy situation and point cloud is more sparse and incomplete. Yet, we will evaluate our models on some other datasets.
>
> For S&W 5:
> * A lot of prior methods were trained and tested on KITTI, but KITTI cannot provide temporal point cloud sequences for generating the dense object. Also, most of them didn’t provide the code for training their model on waymo dataset. The methods we compared are already the recent state-of-the-art methods on 3D object detection with similar experimental sitting.

---

### Official Review · Reviewer_6Pvq · 2022-07-09

**Rating:** 7
**Confidence:** 3
**Soundness:** 4 excellent
**Presentation:** 3 good
**Contribution:** 3 good

**Summary:**

The paper proposes a method to boost the performance of object detectors in 3D LiDAR scans by learning an auxiliary task to densify them (in a self-supervised way, by distilling intermediate dense features and regressing densified point cloud in a DAE fashion). It is fairly detector-agnostic and can thus be applied to different base SOTA detectors such as CentrePoint [1] or SECOND [15]. It improves their performance by 2–5 p.p. at a cost of adding 15–20% to the runtime, which is a reasonable trade-off.

**Questions:**

The only substantial issue to address is missing ablations: L1 and pedestrian category, and skipping predicting the offset.


**Limitations:**

They are addressed in conclusions; I cannot think of anything beyond it that is relevant to the proposed extension.

**Strengths And Weaknesses:**

I think the paper proposes a simple, practical, and universal method, which is sufficiently evaluated, and the paper is well written. I vote to accept it, although I am not very confident as 3D detection is not really my area.

Clarity:
* (+) the paper is generally well written and has sufficient details,
* (−) I think the paper will benefit from explaining the background on how anchor-free and anchor-based methods work; related to that: what is L_reg, specifically? Is it regressing centres of bounding boxes, or corners, or else?
* (=) the hyperparameters are anisotropic, like grid cell size in l. 146; the lateral bounds in l. 230 are 75.2 m which is both large and weirdly specific – is it really so? then why?

Method:
* (+) reasonable design; quite simple – (almost) all added complexity is motivated by ablation,
* (+) the proposed improvement is orthogonal to the design of the underlying detector, so can be universally used;
* (+) if I understand correctly, the densified point cloud is usually obtained from scan sequences, which is probably a common setting, so the method is applicable to a wide range of practical problems,
* (−) PCR module predicts opacities and offsets – if the point cloud in that cell approximates a surface, those offsets are ambiguous in 2 dimensions; is it useful to predict them at all? I suggest to run an ablation where offsets are dropped;
* (=) for PCR: as a future work you can try predicting a different parametrisation like an SDF predicted with an MLP in each cell; it can be naturally supervised with point clouds;

Experiments:
* (+) comparison to SOTA seems sufficient (although I am not an expert);
* (+) ablation study has all necessary baselines (except for dropping offsets in PCR),
* (−) however, ablation results are only reported on 2 out of 6 categories (no pedestrians at all);
* (+) Table 3 shows that the benefit of densification increases with the distance to the object,
* (±) latency analysis in Section 4.6 is nice, however I am not confident 100 samples are sufficient – can you compute confidence intervals?

==========================

Typos / wording:
* l. 18: missing reference,
* l. 40: semantic points – what are these?
* l. 76: should not be there,
* l. 163: what is BEV?
* ll. 207, 267: adopt ← adapt.

---

> ### Author Response · Authors · 2022-08-02
> **Response to Reviewer 6Pvq**
>
> For Clarity 2:
> * Thanks for your careful review. The regress loss L_reg helps predict the height-above-ground, sub-voxel location refinement, 3D size and rotation for anchor-free-based methods following [1] and helps predict the centres and 3D size and rotation for anchor-based methods.
>
> For Clarity 3:
> * We actually followed prior works [1, 3, 27, 28, 29] to set the grid cell size. We will include further details in the revision. The settings of grid size and detect range are based on the LiDAR sensor range and what size of BEV map that we target.
>
> For Method 4:
> * Actually, we predict one offset for each cell, meaning that each cell contains a single point. The ground truth is the average position of the points in each cell.
>
> For method 5:
> * Thanks for your valuable suggestion! We will try it!
>
> For Experiments 3:
> *  Here we show the full ablation results, and we will put the following table in the revision or supplementary material.
> |       | Vehicle-L2 |   Vehicle-L2    | Pedestrain-L2 |   Pedestrain-L2    | Cyclist-L2 |  Cyclist-L2     |
> |-------|:----------:|:-----:|:-------------:|:-----:|:----------:|:-----:|
> |       |     mAP    |  mAPH |      mAP      |  mAPH |     mAP    |  mAPH |
> | Base  |    63.03   | 62.53 |     63.72     | 58.03 |    65.03   | 63.90 |
> |  +dis |    63.84   | 63.32 |     67.04     | 61.21 |    67.59   | 66.44 |
> |  +s2d |    65.75   | 65.22 |     67.62     | 61.65 |    68.50   | 67.34 |
> |  +pcm |    66.12   | 65.58 |     67.47     | 61.59 |    68.69   | 67.54 |
> | - dis |    65.61   | 65.08 |     64.75     | 58.80 |    65.79   | 64.62 |
>
> For Experiments 5:
> * We have evaluated the latency analysis on the whole validation set for CenterPoint baseline, the result is similar to table 8. We will update the table in the revision.
> |      Detectors      | CenterPoint | CenterPoint+S2D |
> |:-------------------:|:-----------:|:---------------:|
> | Inference time (ms) |     53.0    |   62.8 (+9.8)   |

---

### Official Review · Reviewer_oGcw · 2022-07-10

**Rating:** 6
**Confidence:** 5
**Soundness:** 2 fair
**Presentation:** 3 good
**Contribution:** 3 good

**Summary:**

This paper proposes a knowledge-distillation approach to learning densified 3D features for outdoor 3D object detection. Concretely, the authors propose to train two networks DDNet and SDNet for 3D detection. During the training, DDNet will take densified 3D point cloud as inputs~(through multiple frame aggregation) while SDNet only takes single-frame Lidar point cloud. During the training, features computed from SDNet are matched with the corresponding features computed through DDNet at multiple different levels. This feature mimicking encourages the SDNet network to learn densified 3D features even with single frame input. The final model is evaluated on the Waymo 3D Detection and Domain adaptation dataset where the method considerably outperforms multiple popular baselines while maintaining similar latency.

**Questions:**

Q1: Please address the weakness part I listed above. I will adjust my rating mainly based on the author's response to this part.

Q2: In section 3.2, the authors mention that the model filters out the outlier points after aggregating the multi-frame point cloud. How is done exactly?

Q3: line 162: more dense -> denser


**Limitations:**

The authors adequately addressed the limitations and potential societal impact.

**Strengths And Weaknesses:**

S1: The idea of learning densified 3D features is interesting and widely applicable to a wide range of 3D detection problems where the point cloud is sparse due to the physical constraints of the Lidar sensor.

S2: The design of the feature matching process is novel. The feature mimicking or knowledge distillation is performed at multiple levels including latent features and raw point clouds~(through a point cloud completion task).

S3: The method is effective and the experimental validation is solid.

W1: Some important references and discussions with previous work are missing. For instance, [1] also proposed a knowledge distillation approach to learn densified features with single frame input. While it seems that the current method performs considerably better than [1], the authors need to add more detailed discussions to show the difference/improvements.

W2: In section 3.4, the authors claim that they propose a novel voxel-level reconstruction scheme (occupancy + inner voxel offset regression). To my knowledge, this is already explored in [2]. Please add proper references.

W3: All Waymo results are now evaluated with 20% training data + validation set. A comparison with published SOTA methods on the test set (and 100% training data) is needed to verify generalization.

[1] Wang, Yue, et al. "Multi-frame to single-frame: knowledge distillation for 3d object detection." arXiv preprint arXiv:2009.11859 (2020).

[2] Xu, Qiangeng, et al. "Spg: Unsupervised domain adaptation for 3d object detection via semantic point generation." Proceedings of the IEEE/CVF International Conference on Computer Vision. 2021.

---

> ### Author Response · Authors · 2022-08-02
> **Response to Reviewer oGcw**
>
> For W1:
> * Thanks for your suggestions. There are two main advantages of our method compared with [1]. First, [1] only uses five adjacent frames to generate the dense features as the guidance while our method generates a dense object from the whole data sequence. Hence, our method can better simulate the features even for objects that are further away from the sensor. Second, our method further designs the S2D and PCR to better densify the sparse features. We will add the suggested reference and discussion in the main paper.
>
> For W2:
> * Thanks for your information, we will add this suggested reference in the main paper.
>
> For W3:
> * Thanks for your suggestion. As we only have four RTX3090 GPUs, we need nearly a week to train one model on whole training data. Thus, we followed the strategy in OpenPCDet to use 20% training data + validation data. Note that this strategy was also adopted by many papers follow-up works [3, 27, 28, 29]. To evaluate the generalization ability, we further tested our models on the challenge dataset–Waymo Domain Adaptation dataset and the results demonstrate the generalization ability of our model over prior methods. We will be happy to also train our models on the full training data and show the results on the test set in the revised paper.
>
> For Q2:
> * We actually used the function from Open3D, called “remove_radius_outlier”. We will make relevant clarifications in the paper.
>
> For Q3:
> * Thanks for your careful review!

---

> > ### Comment · Reviewer_oGcw · 2022-08-07
> > **Reply to Response**
> >
> > I thank the authors for the detailed reply. I think the paper is now above the acceptance bar and I increase my rating to weak accept.

---

### Official Review · Reviewer_4BxU · 2022-07-12

**Rating:** 6
**Confidence:** 4
**Soundness:** 3 good
**Presentation:** 3 good
**Contribution:** 3 good

**Summary:**

This paper aims to address object detection from LiDAR point clouds. The key idea is to train a densify network that can densify 3D features in a sparse point cloud so that they match the quality of the 3D features trained on dense point clouds. This densify network can be plugged into different object detection networks and enhance their performance - this is demonstrated in experiments.


**Questions:**

Will the code be released after publication?


**Limitations:**

This is a well written paper. Good idea. Good execution and good experiments. I do not see strong limitations.


**Strengths And Weaknesses:**

+ The S2D network is a lightweight, plug and play network, that can enhance the performance of different object detection networks
+ Lots of good design and engineering choices, e.g., voxelizing dense point cloud, design of S2D network, point cloud reconstruction network, etc
+ The experiments are done thoroughly and show improvements on a number of different object detection networks

- Lacks discussion of the reconstruction quality - it would help readers to understand if S2D only learns the feature in latent space, or actually starts to learn 3D dense geometry

---

> ### Author Response · Authors · 2022-08-02
> **Response to Reviewer 4BxU**
>
> Thanks for your suggestions, we have visualized the feature map of S2D in Figure 1 and found that our S2D can learn the 3D dense geometry of the vehicles that are far from the camera. We also visualized the voxel mask and point offset in Figure 5 and found the PCR can really recover some 3D dense geometry information of cars, pedestrians and cyclists. We will provide more visualization in the revision.
>
> Q: Will the code be released after publication?
> - Yes, absolutely!  We will release the code upon the publication of this work.

---

### Author Response · Authors · 2022-08-02
**Response to all reviewers**

Dear Reviewers: Thank you for your effort and time in reviewing our paper. We are encouraged to see positive ratings from all reviewers and your recognition of the good & novel design, interesting idea, effective & universal method, and solid validation.  We will carefully revise the paper following the review comments, include the suggested references, and improve the clarity of the paper.  Also, we will release our code to facilitate future research upon the publication of this work.

---

### Meta-Review · Area_Chair_Pc45 · 2022-08-28

**Recommendation:** Accept
**Confidence:** Certain

**Metareview:**

This paper proposes to utilize point cloud completion tools to densify sparse point clouds which could subsequently improve the performance of point cloud detection methods. After rebuttal, reviewers agree on the novelty of the method and its effectiveness on the Waymo open dataset. AC recommends this paper for acceptance following the unanimous opinion.

**Award:**

No

---

### Decision · Program_Chairs · 2022-09-14

Accept